# Automatic Segmentation of Mandible from Conventional Methods to Deep Learning—A Review

**DOI:** 10.3390/jpm11070629

**Published:** 2021-07-01

**Authors:** Bingjiang Qiu, Hylke van der Wel, Joep Kraeima, Haye Hendrik Glas, Jiapan Guo, Ronald J. H. Borra, Max Johannes Hendrikus Witjes, Peter M. A. van Ooijen

**Affiliations:** 13D Lab, University Medical Center Groningen, University of Groningen, Hanzeplein 1, 9713 GZ Groningen, The Netherlands; b.qiu@umcg.nl (B.Q.); h.van.der.wel@umcg.nl (H.v.d.W.); j.kraeima@umcg.nl (J.K.); h.h.glas@umcg.nl (H.H.G.); m.j.h.witjes@umcg.nl (M.J.H.W.); 2Department of Radiation Oncology, University Medical Center Groningen, University of Groningen, Hanzeplein 1, 9713 GZ Groningen, The Netherlands; p.m.a.van.ooijen@umcg.nl; 3Data Science Center in Health (DASH), University Medical Center Groningen, University of Groningen, Hanzeplein 1, 9713 GZ Groningen, The Netherlands; 4Department of Oral and Maxillofacial Surgery, University Medical Center Groningen, University of Groningen, Hanzeplein 1, 9713 GZ Groningen, The Netherlands; 5Medical Imaging Center (MIC), University Medical Center Groningen, University of Groningen, Hanzeplein 1, 9713 GZ Groningen, The Netherlands; r.j.h.borra@umcg.nl

**Keywords:** mandible segmentation, 3D virtual surgical planning, convolutional neural networks, machine learning

## Abstract

Medical imaging techniques, such as (cone beam) computed tomography and magnetic resonance imaging, have proven to be a valuable component for oral and maxillofacial surgery (OMFS). Accurate segmentation of the mandible from head and neck (H&N) scans is an important step in order to build a personalized 3D digital mandible model for 3D printing and treatment planning of OMFS. Segmented mandible structures are used to effectively visualize the mandible volumes and to evaluate particular mandible properties quantitatively. However, mandible segmentation is always challenging for both clinicians and researchers, due to complex structures and higher attenuation materials, such as teeth (filling) or metal implants that easily lead to high noise and strong artifacts during scanning. Moreover, the size and shape of the mandible vary to a large extent between individuals. Therefore, mandible segmentation is a tedious and time-consuming task and requires adequate training to be performed properly. With the advancement of computer vision approaches, researchers have developed several algorithms to automatically segment the mandible during the last two decades. The objective of this review was to present the available fully (semi)automatic segmentation methods of the mandible published in different scientific articles. This review provides a vivid description of the scientific advancements to clinicians and researchers in this field to help develop novel automatic methods for clinical applications.

## 1. Introduction

Three-dimensional (3D) medical imaging techniques have a fundamental role in the field of oral and maxillofacial surgery (OMFS) [1,2]. 3D images are used to guide diagnosis, assess the severity of disease and for pre-operative planning and per-operative guidance using 3D images and virtual surgical planning (VSP) [3]. In the field of oral cancer, where surgical resection requiring the partial removal of the mandible is a common treatment, resection surgery is often based on 3D VSP to accurately design a resection plan around tumor margins [4]. In orthognathic surgery and dental implant surgery, 3D VSP is also extensively used to precisely guide mandibular surgery [3]. Image segmentation from the radiography images of the head and neck (H&N), which is a process to create a 3D volume of the target tissue, is a useful tool to visualize the mandible and quantify geometric parameters [5]. Studies have shown that 3D VSP requires accurate segmentation of the mandible, which is currently performed by medical technicians [4,6]. Mandible segmentation, especially for 3D VSP, was usually done manually, which is a time-consuming and poorly reproducible process [7,8].

The mandible is located in the lower part of the facial skeleton and consists of the body, rami, angle, and condyles. The different anatomical regions of the mandible have varying densities, which is especially true if the teeth and surrounding soft tissue are also considered, which makes accurate manual segmentation of the mandible time-consuming and challenging [9]. Because a 3D model of the mandible is necessary for 3D VSP in OMFS, and time is often a limiting factor in the clinical workflow, fast mandible segmentation has become a frequent topic of research in recent decades [7]. Researchers have attempted to automate the segmentation process of the mandible, as well as all kinds of tissues from medical images, to reduce the processing time and also the inter-observer variability [10]. Recent advances in image segmentation [11,12,13,14,15] have enabled their applications to medical image segmentation, which stimulate progress in automating mandible segmentation. Although significant progress has been made in mandible segmentation, state-of-the-art methods still experience unsatisfactory outcomes, with several challenges to be solved. For oncology segmentation, the segmentation has to be made for an affected bone (with the tumor), which is a different task from segmenting a bone in implantology or for orthognathic purposes. In implantology, many patients have incomplete teeth, which can accordingly lead to resorption of the bone causing shape variation in the mandible among patients. In orthognathic surgery, the population considered varies greatly in shape in addition to asymmetric, which is also a possible challenge for (automatic) segmentation. Figure 1 shows research challenges in automatic segmentation algorithms for the mandible. The challenges associated with mandibular bone segmentation can be categorized as follows:The presence of anatomically complex bony structures in the scans. As the examples show in Figure 1a, a normal H&N scan includes other bony structures, with a complex anatomy and a similar density. Determining the correct boundaries and separating the mandibular bone from the other bones may be challenging.Artifacts. When X-rays pass through high-density structures or materials, including teeth, postoperative metal implants, etc., the signal on the detectors will change, which will lead to attenuation calculation errors in the (cone beam) computed tomography (CBCT/CT) reconstruction process and consequently cause high noise and strong artifacts in the visual impression of the scans [16]. The mandible boundaries nearby teeth tend to be blurred and hard to detect. In particular, the boundaries of mandible rami are difficult to be identified when dental braces and metal implants badly affect the image quality [17], as shown in Figure 1b. Furthermore, the fact that the superior and the inferior teeth are at the same slice and even overlapping that makes segmentation methods challenging, as shown in Figure 1c.Low contrast. Due to the tomography process and the thinness of condyles, the slices in condyles commonly have low contrast [18], as shown in Figure 1d. Especially in cone-beam CT (CBCT) scans, the condyles in the scans are more blurred than in CT because of its intensity inhomogeneity [19,20].Variation in the appearance. As the examples show in Figure 1e, mandibular shapes vary in shape and size between individuals [18]. Additionally, the shape of the mandibular condyles and body is extremely variable among patients and different age groups [21].Annotation Bias. The manual mandible segmentation often leads to inter-observer variability (Dice score of 94.09% between two clinical experts) [22], which directly influences the quality of treatment planning.

In the present article, a review has been conducted, where the literature available in PubMed and Web of Science databases relating to these studies was considered. In the literature, a variety of segmentation methods, based on statistical shape model, atlas and machine learning, are proposed for head and neck scanning. The selected publications were divided into several categories depending on the type of method used and discussed extensively.

## 2. Method for Literature Selection

In February 2021, a search was conducted on the Web of Science (https://apps.webofknowledge.com (accessed on 23 February 2021)) and PubMed (https://www.ncbi.nlm.nih.gov/pubmed/ (accessed on 23 February 2021)), with the topic keywords and search builder: (automat* AND segment* AND mandible AND 3D). While searching the literature, no specific time line was set. The literature search was performed by the two observers.

In this article, the lists of literature acquired by the above-mentioned database search were imported into Rayyan QCRI [23]. Using the web application, duplicates were searched and removed accordingly. Subsequently, another search was performed to exclude the articles that did not include “segment” or “mandible” keywords in their title and abstracts. Consequently, the irrelevant studies were removed by manually screening the titles and abstracts of journal articles. The obtained relevant publications were further supplemented with selected publications found in their list of references. A detailed analysis of the resulting publications was then conducted from the perspective of image modality, image database, evaluation metrics, and methodology.

## 3. Results

In the field of automatic 3D mandible segmentation, the search on the Web of Science and PubMed databases yielded 33 and 45 results, respectively. After detecting the duplicates in Rayyan, 23 duplicates were removed. After reviewing their abstracts by two observers, 12 were considered to be relevant and were further supplemented with selected publications from their list of references. In total, we collected 77 publications focused on mandible segmentation or organs at risk (OARs) (including mandible) segmentation in the H&N region. Here, we emphasize that we collected 41 publications from the reference of [24]. An overview of the results of the literature search is presented in Figure 2.

### 3.1. Image Modality

CT imaging is primarily performed for mandibular bone segmentation in their works since CT images provide a promising visibility of the bone structures. There are 70, 7 and 4 publications using CT, CBCT and magnetic resonance imaging (MRI), respectively, to visualize H&N (note: some articles applied the scans from multiple image modalities). CBCT images are applied in H&N because of its lower radiation dose, lower cost and faster scanning time than conventional CT [25]. In particular, CBCT is often used for validating patient settings or adjusting treatment plans to adapt to anatomical changes. While CT images provide a good visibility of the bony anatomy, the contrast differences between various soft tissues are much lower than those in MRI. In the consensus on manual delineation of the mandible based on CT, it is also necessary to use MR images, in addition to CT, to help delineate surrounding tumors and soft tissues (note: PET(/CT) imaging technique is also helpful for the tumor delineation [26,27,28]). Furthermore, considering the ionizing radiation of CT/CBCT scanning and the development of computer vision techniques, a strategy of MRI-only treatment planning has become increasingly valuable and feasible in the field of 3D VSP [3].

### 3.2. Image Database

Among the reviewed publications, there are four databases of H&N images that are publicly available. The Public Domain Database for Computational Anatomy (PDDCA) (http://www.imagenglab.com/newsite/pddca/ (accessed on 24 January 2019)) [29], an open-access resource of medical images for cancer research, consists of 48 patient CT images from the Radiation Therapy Oncology Group (RTOG) 0522 study of Head-Neck Cetuximab [30] database of The Cancer Imaging Archive (TCIA) [31], together with manual annotation of the mandible, brainstem, etc. [29]. The organizer of PDDCA provides 40 CT scans with mandible annotations. This dataset has been used for the Head and Neck Auto Segmentation MICCAI Challenge (2015) [29]. According to the protocol of the Challenge, 25 of 40 scans (0522c0001-0522c0328) are employed as the training dataset, and the other 15 scans (0522c0555-0522c0878) are employed as the test dataset [29]. ‘Organ-at-risk segmentation from head & neck CT scans’ is one of four tasks in StructSeg 2019 (https://structseg2019.grand-challenge.org/Dataset/ (accessed on 23 February 2021)), consisting of 50 nasopharynx cancer patient CT scans with several manually delineated organs at risk including mandibles (left and right). The StructSeg 2019 dataset has been used for the Automatic Structure Segmentation for Radiotherapy Planning Challenge 2019. Each of the annotated CT scans is marked by one experienced oncologist and verified by another experienced oncologist.

Furthermore, a dataset has been augmented or combined into new publicly available databases by Tang et al. [32] in 2019, for example, the manual delineations of 28 OARs in 35 CT scans from the Head-Neck Cetuximab and 105 CT scans from Head-Neck-PET-CT [33] databases (https://github.com/uci-cbcl/UaNet#Data (accessed on 23 February 2021)). A subset of 31 CT scans, which was used for the test and validation set in the study from Nikolov et al. [24], was collected from two datasets of The Cancer Genome Atlas Head-Neck Squamous Cell Carcinoma [34] and Head-Neck Cetuximab [30] from TCIA. The subset with the ground truth added is available at https://github.com/deepmind/tcia-ct-scan-dataset (accessed on 23 February 2021).

### 3.3. Evaluation Metrics

In addition to the differences in the used dataset, the results are evaluated and presented in different ways in the reviewed papers. Moreover, there are no standard metrics for the segmentation evaluation, so different evaluation metrics are used to report the segmentation performance. For segmentation, the evaluation metrics are mainly divided into three categories: overlap-based metrics, distance-based metrics, and volume-based metrics. The metrics are summarized in Table 1. The overlap-based metrics indicate the difference in overlap measurement between automatic prediction and manual segmentation, which can be obtained by four indicators: true positive (TP), false positive (FP), false negative (FN) and true negative (TN). The TP value is the number of pixels segmented correctly as foreground. The count of pixels falsely classified as the foreground is given by the FP value. The total count of falsely classified as background pixels is represented by FN. The TN value represents the correctly classified background pixels. The most commonly used overlap-based metrics include the Dice similarity coefficient (Dice), Sensitivity (Sen), false positive volume fraction (FPVF) [35], false negative volume fraction (FNVF) [35], etc. To measure the contour difference between automatic and manual segmentation, the most commonly used metrics are distance-based metrics. In the context of mandibular segmentation, the following distance-based metrics have been frequently used: average symmetric surface distance (ASD), Hausdorff distance (HD), 95th-percentile Hausdorff distance (95HD), mean square error (MSE) [36], and root mean square error (RMSE) [36]. In some medical image segmentation tasks, the volume of the object is also very important for treatment planning, and the metric based on volume is helpful to evaluate the performance of the segmentation method. Volume overlap error and volume error are the common indices to evaluate the results of mandibular segmentation.

### 3.4. Methodology

In this paper, the mandible segmentation approaches have been broadly divided into six categories, namely, (1) statistical shape model-based (SSM-based), (2) active shape model-based (ASM-based), (3) active appearance model-based (AAM-based), (4) atlas-based, (5) level set-based, (6) classical machine learning-based and (7) deep learning-based approaches, as depicted in Table 2. There are several hybrid segmentation methods proposed in the collected literature. The hybrid methods are introduced in the listed seven categories.

#### 3.4.1. SSM-Based, ASM-Based and AAM-Based Methods

Statistical shape model-based (SSM-based) segmentation algorithms take advantage of the prior shape information to extract the structures of the objects. A dataset of object shapes is needed for training in this technique. Procrustes alignment is applied to align the landmark point set of the objects, which is placed at key features and/or along the contour of the object shapes [107]. Principal component analysis (PCA) is employed to build a template shape of the objects [108].

The active shape model (ASM) [109] is the one of the commonly used image segmentation techniques based on the SSM approach. In 2006, Lamecker et al. [40] used a vanilla SSM-based segmentation method to extract the anatomical variability of developed mandible shapes in the training stage and match the statistical mandible shape to a given CT dataset via a deformable model approach in the testing stage. Kainmueller et al. [42] developed a segmentation method based on an SSM and a Dijkstra-based optimization for reconstructing the mandible including the course of the alveolar nerve. Albrecht et al. [41] adopted a multiatlas-based segmentation to obtain the coarse segmetnation for the OARs and then employed the active shape model (ASM) to finely segment the OARs based on the results from coarse segmentation.

The active appearance model (AAM) [110] is an extension of SSM and ASM to further statistically model the texture information of the object. Two statistical models of shape and texture are further merged into an appearance model [110]. Babalola et al. [44] applied the AAM approach to automatic mandible and brainstem segmentation. Mannion- Haworth et al. [43] utilized the groupwise image registration method-based minimum description length approach [111] and AAM [110] built from manually segmented objects in CT images.

Moreover, several publications on SSM-based segmentation methods have been used in combination with other strategies to enhance the segmentation outcomes. Hybrid methods have demonstrated good results [18,37,38,39]. Kainmueller et al. [39] used SSM adaptation for mandible segmentation and then utilized the graph cut algorithm for fine segmentation in the MICCAI 2019 challenge [112]. Gollmer et al. [37] applied the SSM method with optimized correspondence for mandible segmentation. The authors established correspondence by optimizing a model-based cost function. Moreover, the authors introduced a relaxed SSM method for mandible segmentation [38]. Abdolali et al. [18] proposed a framework based on SSM and fast marching for automatic segmentation of the mandibular bone and canal. Moreover, the authors utilized low-rank decomposition for preprocessing. Table 3 presents the segmentation performance using SSM-based, ASM-based and AAM-based approaches.

#### 3.4.2. Atlas-Based Methods

Atlas-based segmentation methods first utilize deformable registration approaches to register a known reference segmentation mask (that form an atlas) to a patient [50]. The registration optimization problem is usually solved by searching the deformation space. Then, this deformation is applied to contours made on the atlas to project the contours back to the patient-space.

In 2007, Zhang et al. [50] used an atlas-based image segmentation method to automatically segment ROIs in H&N CT images. Chuang et al. [45] presented an atlas-based semiautomatic segmentation for the mandible. The authors firstly cropped a minimum enclosing box of the mandible in a raw CT image followed by a user-determined global threshold to remove nonosseous tissue and then used the output from these registrations to transform the 3D mandible template model of the respective template scans into a mandible model that maps to the input test scan. In another paper [62], a semisupervised registration algorithm based on the atlas was developed to accurately segment the OARs with a real ground contour and all other coarse OARs in the atlas. The method concatenates rigid and deformable blocks, takes an atlas image, a group of atlas-space segmentations and a patient image as inputs and outputs the patient-space segmentation of all OARs defined on the atlas. Qazi et al. [51] combined atlas registration and organ-specific model-based segmentation in a common framework to segment various organs at risk in H&N CT images. The authors applied atlas registration and organ-specific model-based segmentation at a global level and then used a probabilistic refinement step to refine at the voxel level. Ayyalusamy et al. [54] applied an atlas-based method for OAR segmentation in H&N and analyzed the dependence of atlas-based automatic segmentation for different levels on anatomy matching between sample patients and atlas patients.

In the case of mandible segmentation, the atlas-based method has become increasingly popular, so it has been frequently implemented in commercial or open-source tools. There are several articles using an atlas-based tool, for instance, Advanced Medical Imaging Registration Engine (ADMIRE) v1.05 (Elekta Software) used by [56], Smart Probabilistic Image Contouring Engine (SPICE) used by [57,58], and PLASTIMATCH MABS provided by [61]. Moreover, La Macchia et al. [60] quantitatively analyzed three different automatic atlas-based segmentation software offerings for adaptive radiotherapy in a small dataset.

In regard to the fusion strategy, Mencarelli et al. [46] built a knowledge base and thresholding technique for nine high-contrast structures (including the mandible) for an online or offline image-guided radiotherapy (RT) application. Then, they further used a hidden Markov model to identify the structures. Wang et al. [49] developed a mandible and maxilla segmentation approach in an H&N CBCB scan that estimated a patient-specific atlas applying a sparse label fusion strategy from predefined CT atlases and then converted it into a convex optimization problem using maximum a posteriori probability for mandible/maxilla segmentation. Gorthi et al. [52] developed a novel segmentation method that used active contour-joint registration and an atlas selection strategy.

A multiple atlas strategy or its modified version is also commonly used because the propagated label cannot be generalized in conventional atlas-based models. To address this issue, multiatlas approaches have been proposed. Chen et al. [47] presented a multiatlas-based method for multiple structure segmentation in CT scans. The authors registered CT images with the atlases at the global level so that structures of interest are aligned approximately in space. Based on that approach, the multiatlas-based segmentation can be performed at the local level. A method presented by Haq et al. [55] dynamically selected and weighted the proper number of atlases for weighted label fusion and predicted the segmentations and consensus maps, indicating voxel-wise agreement between different atlases that were selected from those exceeding an alignment weight called the dynamic atlas attention index. Alignment weights were computed at the image level (called global weighted voting) or at the structure level (called structure weighted voting). McCarroll et al. [59] studied an automatic segmentation strategy for H&N cancer to better provide a fully automated radiation treatment planning solution for low- and middle-income countries using an atlas-based deformable-image-registration algorithm. Han et al. [48,53] adopted a novel hierarchical multiatlas registration approach for OAR segmentation. Table 4 shows the summary of atlas-based method used.

#### 3.4.3. Level Set-Based Methods

The level set-based image segmentation algorithm is modified from the snake algorithm [113]. It requires an initial contour curve, and then, the curve evolution is performed via minimizing the functional energy. To obtain a better prediction, the studies [9,63] make use of the atlas method to generate an initial contour in their segmentation tasks. Patch-based sparse representation has been applied to estimate the patient-specific atlas and then combined with a level set framework using the rule of maximizing a posteriori probability. With these methods, the mandible segmentation problem can be converted into a convex optimization problem [9]. Table 5 shows a summary of the level set-based method used.

#### 3.4.4. Classical Machine Learning-Based Methods

In the past few decades, the role of machine learning in medical applications has greatly increased. Classical machine learning (CML) methods include thresholding techniques, linear regression, support vector machines, random forests, etc. Many studies have used CML methods to segment the mandible, as listed in Table 6.

Normally, this technique is implemented in a hybrid way. A multiple threshold method by Otsu [114] was used to calculate proper thresholding values in the segmentation phase in the article from Barandiaran et al. [67]. After thresholding, region growing was applied so that many bone structures could be segmented, such as parts of the jaw bone or spine. The largest connected component of the segmented image volume can be simply selected for mandible segmentation [67]. Later, Wang et al. [65] made use of random forests to segment the mandible and maxilla in their expended CBCT database and achieved better results than their previous method [9]. Linares et al. [19] proposed a semiautomatic two-stage segmentation method for CBCT. The authors first performed bone segmentation using a supervoxel technique and then implemented graph clustering with an interactive user-placed seed process that was used for mandible and skull segmentation. In this way, supervoxel methods can reduce the excessive number of voxels of CBCT 3D volume, leading to shorter processing time. Orbes-Arteaga et al. [68] introduced a new patch-based label fusion approach to weight the label votes using a generative probabilistic approach, in which local classifiers from the dictionary of atlas patches were generated for weighting the probability of the target. Wang et al. [69] integrated shape priors with a hierarchical learning model, in which they introduced 3 novel strategies: hierarchical critical model vertex identification, joint learning of shape and appearance, and hierarchical vertex regression. Qazi et al. [70] applied point-based registration and model-based segmentation and introduced a novel voxel classification approach to improve the results from model-based segmentation. Torosdagli et al. [71] adopted random forest regression to detect the mandible in 3D and then utilized a 3D gradient-based fuzzy connectedness image segmentation algorithm for final segmentation. Wu et al. [72] built population fuzzy anatomy models to find the hierarchical relationship of the objects. Then, the authors further utilized the anatomical information in the hierarchical relationship to locate the objects. Finally, the segmentation could be generated according to the location results. Tam et al. [36] presented support vector regression to predict all the boundary points of the organs of interest. Gacha et al. [75] performed histogram equalization and grayscale morphology for mandible segmentation. Then, image binarization was performed after grayscale method. Finally, 3D template matching was performed using cross-correlation to find a point inside the mandible followed by a geodesic dilation using the binary image as mask. Tong et al. [73] introduced new features into the automatic anatomy recognition approach from [66] in which they combined texture and intensity information into the recognition procedure. Wu et al. [74] developed a methodology, called AAR-RT, which extended their previous AAR framework [66] to RT planning in H&N. Spampinato et al. [76] applied a fully automatic mandible segmentation method to support orthodontists in assessing facial asymmetry, in which they used morphological operations for extraction of the mandible, and then, the connected components were computed and considered part of the mandible in each 2D slice.

Along with the quick development of computer and MRI imaging technology, MRI images have also been used in mandible segmentation. Ji et al. [64] presented a two-stage rule-constrained seedless region growing approach for the mandible segmentation in MRI images. A 3D/2D seedless region growing approach is used to detect a trabecular bone and cortical bone of mandible after using a thresholding technique. The mandibular body was finally merged with some morphological processes. Udupa et al. [66] developed an automatic anatomy recognition methodology in H&N MRI scans that is based on fuzzy modeling ideas and tightly integrated fuzzy models with the iterative relative fuzzy connectedness delineation algorithm to locate and delineate organs in different body scans.

#### 3.4.5. Deep Learning-Based Methods

In recent years, deep learning-based segmentation algorithms have become popular among researchers. With the development of deep learning technology, deep learning methods have shown a tremendous performance in the area of image segmentation. Deep learning techniques provide more flexibility and powerful capabilities than the traditional machine learning methods and require less expert analysis, facilitating extension to the other segmentation tasks [115]. For instance, the popular deep learning architectures, including UNet [13], SegNet [12], etc., are widely used for automatic image segmentation. Consequently, the deep learning approach has shown promising applications in the problem of mandible segmentation. Over the years, many studies have provided better solutions using deep learning architectures than other classical image segmentation methods, as shown in Table 7, Table 8, Table 9, Table 10 and Table 11.

Ibragimov et al. [77] proposed the first attempt of using the CNN-based concept to segment OARs in H&N CT scans. Furthermore, the authors smoothed the obtained segmentation results through a Markov random fields algorithm. To address the challenge of low performance on small low-contrast structures in CT, a new loss function, namely, the batch soft Dice loss function, was developed by Kodym et al. [78]. The authors used the new loss function in UNet and achieved better results than the methods using other loss functions. Yan et al. [79] developed a symmetric convolutional neural network to force convolution and deconvolution computation to be symmetric. Table 7 shows the summary of using deep learning-based methods.

Another intuitive method of 3D segmentation is to train a 3D network to process volume data directly. In this way, the 3D segmentation network can extract 3D structure information. AnatomyNet [82] is built in the 3D U-net architecture using residual blocks in encoding layers. Moreover, a new loss function combining the Dice score and focal loss is applied in the training process. Xia et al. [83] presented a methodology to automatically measure the masseter thickness and locate the ideal injection point for botulinum toxin into the masseter from a CT scan, in which a 3D UNet with a Resblock is used for the mandible and masseter segmentation. Willems et al. [84] applied the 3D segmentation network from [116]. The method consists of a 3D CNN and fully connected 3D CRF. Ren et al. [85] introduced multiscale patches as input for representing the center voxel and designed a 3D CNN network for segmentation of OARs. In addition, the authors interleaved the CNNs designated for the individual tissues since the neighboring tissues are often related on the anatomical side. With this strategy, the patch segmentation result of a specific tissue can be refined by the other neighboring tissues. This study can deal with large tissues, while the small tissues are still not that promising in terms of Dice and 95HD. For contending with the imbalance between classes mainly caused by small organs, He et al. [86] proposed a combined UNet model that consists of 2D UNet, 3D UNet and 3D small UNet. The first model is a 2D UNet model, which has advantages in processing thick slice images. The second model is a 3D UNet model, which can cover most organs with the original resolution in the transverse plane via clipping the scans. The third model is a 3D small UNet model, which focuses on the segmentation of small organs, clipping from the boundary box of the 2D UNet model. Rhee et al. [87] proposed a CNN-based autocontouring tool that can be used to detect the errors in autocontours from a clinically validated atlas-based autocontouring tool. They used the DeepMind model [24] to detect the ill-defined contours from an atlas-based autocontouring tool. Nikolov et al. [24] applied a 3D UNet architecture for OAR segmentation and demonstrated that the 3D UNet approach achieved expert-level performance in delineating multiple organ segmentation in the H&N region. Tong et al. [88] proposed a novel deep learning-based approach based on a generative adversarial network (GAN) with a shape constraint (SC-GAN), where a DenseNet was used to predict the segmentation and a CNN-based discriminator network was applied to correct the predicted errors and image-level inconsistency between the predicted image and ground truth. Moreover, a shape representation loss was applied into the segmentation and adversarial loss function for reducing the false positives. Chan et al. [81] provided a lifelong learning protocol to improve the prediction accuracy of OAR segmentation, in which they used a multitask learning scheme coupled with transfer learning to accomplish this. In this way, the network transfers its shared knowledge to single tasks to help improve the generalizability of the network on limited datasets. To cope with the lack of overall shape and smoothness of OARs, Xue et al. [89] developed the feasibility of learning the signed distance map (SDM) directly from medical scans, in which SDM is usually calculated from object boundary contours and the binary segmentation map. To utilize more information in traditional segmentation training, the authors introduced an approximated Heaviside function to train the model by predicting SDMs and segmentation maps simultaneously. Table 8 lists the summary of literature that used 3D networks for mandible segmentation.

However, 3D networks require clipping or resampling the input data into small 3D patches to reduce GPU memory [92]. In addition, integration of 3D patches in post-precessing is a time-consuming workflow. Therefore, to overcome the drawback of 3D networks, using a 2.5D or multiview training strategy instead of 3D is a potential strategy [5,90]. Qiu et al. [5] adopted 2.5D volume of CTs as input in UNet and then combined the resulting 2D segmentations from three orthogonal planes (axial, sagittal and coronal) into a 3D segmentation. This architecture can take into account the spatial information of adjacent slices in order to preserve the connectivity of anatomical structures. Lei et al. [90] first presented a segmental linear function to enhance the intensity of CT images, which makes the organs more separable than the existing simple window width method. Then, the authors proposed a novel 2.5D network for accurate OAR segmentation. Additionally, they applied a novel hardness-aware loss function to make the network give more attention to hard voxels. Liang et al. [91] developed a novel multiview (i.e., axial, coronal, and sagittal view) spatial aggregation framework for joint localization and segmentation of multiple OARs. Additionally, the authors proposed a region-of-interest-based fine-grained representation CNN for predicting the probability maps of OARs for each 2D view of CT images. In this way, the approach unifies the OAR localization and segmentation tasks and trains them in an end-to-end fashion. Qiu et al. [92] integrated a recurrent unit and a vanilla network, which enable the network to learn the continuity of neighborhood slices for the scans in the 2D architecture. Table 9 shows the summary of using 2.5D networks for mandible segmentation.

The attention model has been widely used in various fields of deep learning in recent years [117,118,119]. The attention mechanism focuses limited attention on key information to save resources and obtain the most effective information quickly [118]. Gou et al. [93] designed a self-channel-and-spatial-attention neural network for H&N OAR segmentation, in which spatial and channelwise attention learning mechanisms can adaptively force the network to emphasize the meaningful features and weaken the irrelevant features at the same time. A novel simple local blockwise self-attention-based segmentation approach was presented by Jiang et al. [94]. This approach can more easily make the network learn the spatial location and interrelation within input images. This study demonstrated that adding the additional attention blocks increases the contextual field and captures focused attention from anatomical structures. Sun et al. [95] presented an end-to-end CNN, called AttentionAnatomy, which was trained with three partially annotated datasets to segment OARs from the whole body. AttentionAnatomy retains the basis structure of UNet and has two branches: a CT region classification path and an OAR segmentation path. The CT region classification path outputs region prediction and attention vectors for OARs, which represent the inference of possible combinations of OARs in the current image. Then, the OAR segmentation path applies these attention vectors to modulate the final output mask. The authors further proposed a recalibration mechanism to solve the partial annotation problem. Moreover, a hybrid loss function composed of batch Dice loss and spatially balanced focal loss is used to address the extreme category imbalance. Liu et al. [96] developed a cross-layer spatial attention map fusion network to combine different spatial attention maps and establish connections for significant features in the feature maps. Furthermore, the authors adopted a top-k exponential logarithmic Dice loss for the imbalanced dataset in their segmentation tasks. Table 10 shows the summary of using attention strategies for mandible segmentation.

There are several two-stage training strategies, such as cascaded CNNs [97,98,99,105], the combination [101,103] of Faster R-CNN [120] and CNNs. Zhang et al. [97] developed a cascaded CNN architecture for multitasks segmentation in H&N scans, in which a slice classification network is proposed to classify CT slices into the corresponding target categories. Then, the irrelevant slices without the targets were excluded. Next, the slices in the corresponding categories were pushed to a refined 3D segmentation network for target segmentation. In this case, the prediction performance was further improved with the help of the slice classification network. Tappeiner et al. [98] trained two hierarchical 3D neural networks to segment multiple OARs in H&N CT scans. The authors first implemented a coarse network on size-reduced scans for locating the OARs. Then, a subsequent fine network on the original resolution scans was trained for a final accurate segmentation based on the results from the coarse network. Mu et al. [99] applied two cascade networks for location and fine segmentation of organ segmentation, in which the network is modified based on the squeeze and exception module, the residual module and V-Net [121]. There are still some shortcomings in this research. The network input is accomplished by resampling the input images with the same voxel size. However, the difference of the thickness and the resolution within the slices is often too large. Therefore, the operation of resampling in the first network can easily cause localization failure when organs have small volumes. Wang et al. [100] proposed a framework similar to [99]. Tang et al. [32] presented a novel network U_*a*_-Net which contains detection and segmentation stages. The approximate location of the OARs can be identified in the detection stage, and then, fine segmentation can be further performed utilizing the results from the detection stage as a guide in the fine stage. Lei et al. [101,102] adopted a 3D Faster R-CNN to locate the H&N OARs in MRI and CT images and then applied an attention UNet to segment the OARs based on the location results. Liang et al. [103] developed an automatic detection-segmentation network for OAR segmentation in H&N CT scans, in which the authors applied a Faster R-CNN [120] to locate the OARs and a fully convolutional network (FCN) [11] to further predict OAR masks based on the organ bounding boxes from the detection stage. Dijk et al. [104] applied a deep learning-based method from Mirada for OAR segmentation in H&N, in which the method was applied in the AAPM Challenge 2017 [122]. The method first used a CNN network to predict all OARs at a coarse resolution and then applied a series of organ-specific 10-layer networks for each organ segmentation using the predicted coarse results with full resolution images as input. Men et al. [105] developed a cascade CNN for the OAR delineation for radiotherapy in head-and-neck squamous cell carcinoma (HNSCC) of TCIA [33]. The cascaded CNN contains two stages (i.e., a coarse detection stage and a fine segmentation stage). The authors first used a shallow network as coarse detection to find the ROIs. In the fine segmentation stage, they used the coarse prediction as input for fine segmentation. Egger et al. [106] utilized a VGG network to classify the slices where the mandible appears and then employed an FCN network to delineate the mandible. Table 11 shows the summary of using two-stage strategies for mandible segmentation.

A new automatic segmentation algorithm was presented by Xue et al. [80] that employs a novel hybrid neural-like P system to alleviate the challenges in tasks of segmentation of H&N scans. The new P system has the common advantages of cell-like and neural-like P systems so that it can solve more practical problems in parallel. In the new P system, the effective integrated CNNs are implemented at the same time through different initializations to perform the pixel-level segmentation of OARs in order to obtain more effective features and take advantage of the strength of ensemble learning.

## 4. Discussion

Numerous studies have been published regarding mandible segmentation. Mandible segmentation methods collected in this paper are classified into seven categories. As shown in Figure 3a, atlas-based (23.4%), CML-based (19.5%) and DL-based (42.9%) methods are frequently used and developed in mandible segmentation. More than three-quarters of the publications (85.8%) utilized these methods. The atlas-based method is the oldest technique in the field of mandible segmentation and has been frequently implemented in commercial software [56,57,58,60,61]. Low accuracy is reported by using this kind of method due to the image artifacts and low contract. Atlas construction is a time-consuming task when conducting complex nonrigid registration. An extended approach is to apply level set methods that help the initial curve move towards the boundary of the object [9,63]. The level set function implicitly represents the boundary of a curve or a shape through its zero horizontal tangent plane. The main drawback of using the level set-based methods is the requirement to initialize the contour. Thus, using the contours generated from atlas-based methods as initialization is helpful in this task [9,63]. The SSM-based method is another approach used by various researchers for mandible segmentation, in which the shape information of the object is taken into consideration in ASM. The method performed well compared to the atlas-based method. In addition to considering the shape information, AAM considers the texture information of the image and learns it in an integrated statistical shape model.

The most frequently used image segmentation approach is machine learning, especially deep learning (DL), which produces better results than the other methods. The classical machine learning algorithms demand a considerable human effort in feature extraction. The DL technique has been used frequently since 2018 due to the rapid development of DL techniques, as illustrated in Figure 3b. Deep learning methods have resolved the issue of automatically extracting the features. The major issue of using deep learning methods is the requirement of a substantial quantity of annotated medical data, which limits the use of supervised learning techniques and avoids overfitting [93]. Furthermore, to increase the size of training data, the deep learning-based methods are trained using data augmentation schemes such as scaling, cropping, affine transformation, elastic deformation, rotations, and noise contamination [81,82,93,98]. Moreover, with the evolution of deep learning, mandible segmentation has achieved remarkable performance compared to classical segmentation approaches.

As shown in Figure 3c, most of the publications (86.4%) used the CT imaging modality. Very few publications worked with CBCT (8.6%) and MRI (4.9%). Moreover, most of the publications worked with only one single imaging modality. There are 4 publications that worked with two imaging modalities. As shown in Figure 3c, number of publications using CT modality is much more than that of using CBCT and MRI techniques, which need more effort for manual expert annotation. However, the CBCT and MRI imaging techniques involve very low-dose radiation or no radiation. Mandible segmentation in CBCT and MRI is worthy of further exploration. The ability to derive the exact shape of the mandible from MRI’s nonionizing radiation imaging mode provides clinicians with a value-added visualization option to view hard and soft tissues in a visual environment. Especially when high-resolution bone information is not needed, the spatial relationship between soft tissue muscle and the mandibular body can be obtained without additional volume imaging using CBCT. Patients who receive MR imaging but do not require facial CT imaging can benefit from this approach without the need for CBCT or X-ray radiation from CT imaging. This application eliminates the radiation risk to patients [3].

The Head and Neck Auto Segmentation MICCAI Challenge (2015) [29], which used the PDDCA dataset in the challenge, has obtained much attention during the past years. There are 25 publications for mandible segmentation in PDDCA [29]. Among all the metrics summaries, the Dice score is the most popular metric used to evaluate the performance of the segmentation methods. The distribution regarding the publication with the PDDCA test is presented in Figure 3d. The Dice score increases with the year. Although it is a common metric used in image segmentation, it may not be the most relevant metric for clinical applications [82]. Identifying a new metric in consultation with the physicians practicing in the field would be an important next step for real clinical applications of the method [82].

## 5. Conclusions

Mandible segmentation is a challenging task due to the presence of metal artifacts, the low contrast of condyles and the large variation in mandibles among patients. CT modality is the most frequently used imaging technique for OMFS. To date, mandible segmentation methods proposed in the literature have achieved promising results in CT scans. Before the machine learning era, the atlas-based segmentation method was the most common method and performed well. With the advancement of technology, deep learning-based segmentation methods have increased substantially. Deep learning-based methods perform better and much faster than traditional methods, although they require more data and higher computing resources in training. There is still room for improvement since the datasets are limited and cannot fully represent the general patient population in clinic. Furthermore, mandible segmentation in CBCT and MRI is valuable for further exploration due to their low-dose radiation or no radiation. Therefore, mandible segmentation is still an open research area to be improved.

## Figures and Tables

**Figure 1 jpm-11-00629-f001:**
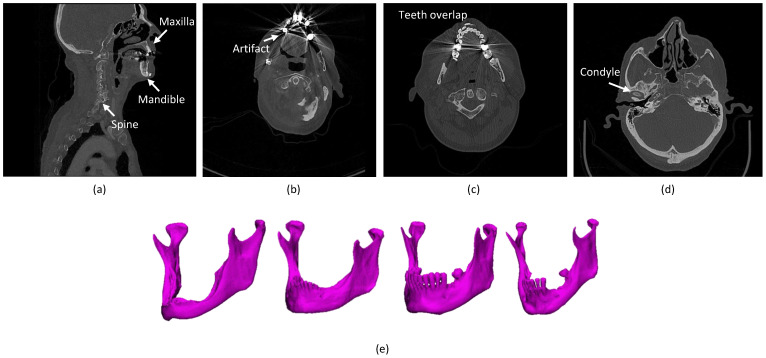
Examples of typical cases that challenge accurate mandible segmentation. (**a**) Various bone-structured organs in the H&N scans. (**b**) Metal artifacts. (**c**) Presence of inferior and superior teeth in the same slice. (**d**) Lower intensity in the condyles. (**e**) Large variation in mandibles between patients.

**Figure 2 jpm-11-00629-f002:**
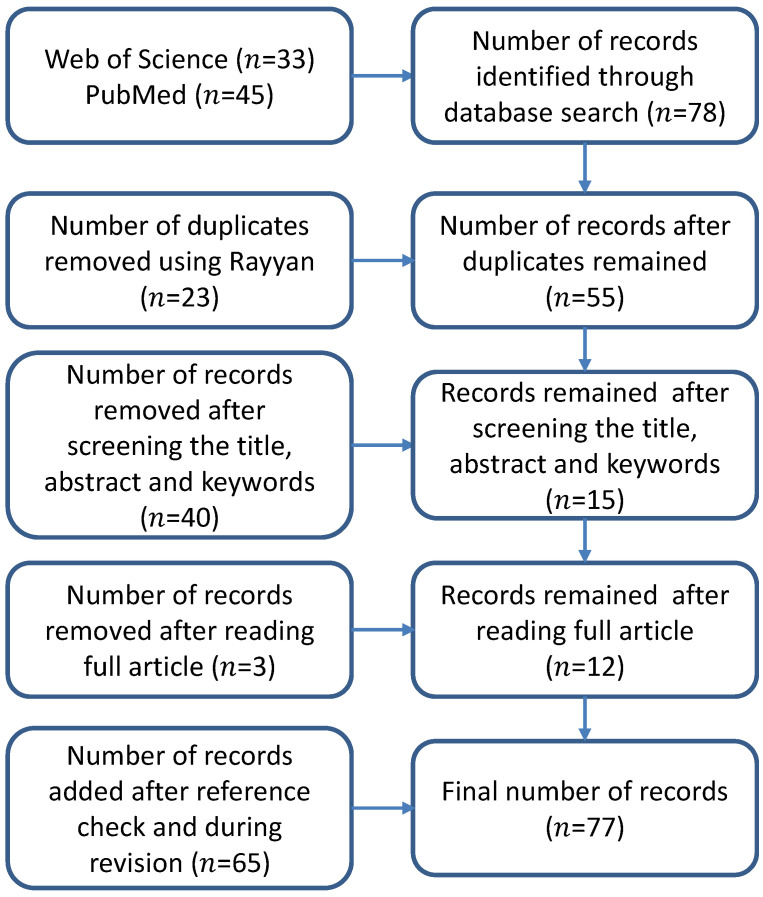
Flowchart of the literature review article selection process.

**Figure 3 jpm-11-00629-f003:**
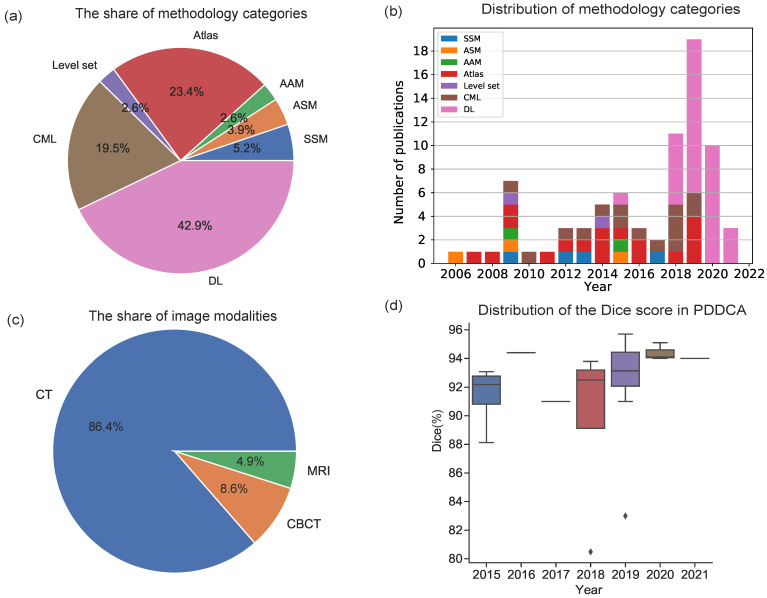
Analysis of publications pertaining to mandible segmentation. (**a**) The share of methodology categories used in the publications. (**b**) Distribution of the publications by year. (**c**) The share of image methodologies used in the publications. (**d**) Box plot of the distribution of the Dice score of the PDDCA test from the publications. The black lines in the middle of the box indicate the average Dice score.

**Table 1 jpm-11-00629-t001:** Performance metrics applied to performance measurement of automatic segmentation of the mandible and the corresponding references and mathematical definitions. Note: $Y_{r}$ indicates the pixels in the reference standard (ground truth), and $Y_{p}$ is the pixels in the automatic segmentation. |.| represents the number of voxels. ∥.∥ represents the $L_{2}$ norm. *a* and *b* are corresponding points on the boundary of *A* and *B*.

Metric	Abbreviation	Definition
Overlap-based metrics, reported in percents (%)		
Dice similarity index	Dice	$Dice=\frac{2|Y_{r}\cap Y_{p}|}{|Y_{r}\left|+\right|Y_{p}|}=\frac{2TP}{2TP+FP+FN}$
Sensitivity	Sen	$Sen=\frac{|Y_{r}\cap Y_{p}|}{|Y_{r}|}=\frac{TP}{TP+FN}$
Recall	Rec	Recall=Sen
Positive Predictive Value	PVV	$PVV=\frac{|Y_{r}\cap Y_{p}|}{|Y_{p}|}=\frac{TP}{TP+FP}$
Jaccard similarity coefficient	Jac	$Jac=\frac{|Y_{r}\cap Y_{p}|}{|Y_{r}\cup Y_{p}|}=\frac{TP}{TP+FP+FN}$
Intersection over union	IoU	IoU=Jac
Specificity	Spe	$Specificity=\frac{|\left(1-Y_{p}\right)\cap \left(1-Y_{r}\right)|}{|Y_{p}|}=\frac{TN}{TN+FP}$
False positive volume fraction	FPVF	$FPVF=\frac{|Y_{p}-Y_{r}|}{|Y_{r}|}=\frac{FP}{TP+FN}$
False negative volume fraction	FNVF	$FNVF=\frac{|Y_{r}-Y_{p}|}{|Y_{r}|}=\frac{FN}{TP+FN}$
Distance-based metrics, reported in millimeters (mm)		
Average symmetric surface distance	ASD	$\mathrm{ASD}\left(A,B\right)=\frac{d(A,B)+d(B,A)}{2},$ where $d\left(A,B\right)=\frac{1}{N}\sum_{a\in A}\min_{b\in B}∥a-b∥,$
Hausdorff distance	HD	$D_{\mathrm{HD}}\left(A,B\right)=\max\left(h\left(A,B\right),h\left(B,A\right)\right),$ where $h\left(A,B\right)=\max_{a\in A}\min_{b\in B}∥a-b∥$
95th-percentile Hausdorff distance	95HD	$95\mathrm{HD}=\max(h^{95\%}\left(A,B\right),h^{95\%}\left(B,A\right)),$ where $h^{95\%}\left(A,B\right)=\max_{a\in A}\min_{b\in B^{95\%}}∥a-b∥$
Mean square error	MSE	$MSE=\frac{1}{n}\sum_{i=1}^{n}{∥A_{i}-B_{i}∥}^{2}$, where $A_{i}$ is the boundary of the *i*-th OAR and $B_{i}$ is the boundary of the *i*-th prediction.
Root mean square error	RMSE	$RMSE=\sqrt{MSE}$
Volume-based metrics, reported in percents (%)		
Volume overlap error	VOE	$1-\frac{|Y_{r}\cap Y_{p}|}{|Y_{r}\cup Y_{p}|}$
Volume error	VE	$VE=\frac{Y_{r}-Y_{p}}{Y_{r}}\;\mathrm{or}\;\frac{Y_{p}-Y_{r}}{Y_{r}}$

**Table 2 jpm-11-00629-t002:** Methodology applied to automatic mandible segmentation in the head and neck region, and the corresponding references.

Methodology Categories	Publications	Number of Publications
SSM-based	[18,37,38,39]	4
ASM-based	[40,41,42]	3
AAM-based	[43,44]	2
Atlas-based	[45,46,47,48,49,50,51,52,53,54,55,56,57,58,59,60,61,62]	18
Level set-based	[9,63]	2
Classical machine learning-based	[19,36,64,65,66,67,68,69,70,71,72,73,74,75,76]	15
Deep learning-based	[5,24,32,77,78,79,80,81,82,83,84,85,86,87,88,89,90,91,92,93,94,95,96,97,98,99,100,101,102,103,104,105,106]	33

**Table 3 jpm-11-00629-t003:** Summary of SSM-based, ASM-based and AAM-based methods.

Study	Year	Datasets	No. of Patients	Performance					Image Modalities	Time/ Equipment	Category
				Dice (%)	ASD (mm)	HD (mm)	95HD (mm)	VOE (%)			
Abdolali [18]	2017	In-house	120	91.38±2.06	0.71±0.09	—	—	—	CBCT	5 min/CPU	SSM-based
Gollmer [37]	2012	In-house	30	—	0.50±0.00	11.30±3.50	—	14.0±1.9	CT	—	SSM-based
Gollmer [38]	2013	In-house	30 + 6	—	0.40±0.08	10.26±3.16	—	11.0±2.0	CT(train)/ CBCT(test)	—	SSM-based
Kainmueller [39]	2009	MICCAI 2009	18	88.40	—	8.40	—	—	CT	15 min/CPU	SSM-based
Lamecker [40]	2006	In-house	15	—	—	—	—	—	CT	—	ASM-based
Albrecht [41]	2015	PDDCA	40	88.13±5.55	—	—	2.83±1.18	—	CT	5 min/CPU	ASM-based
Kainmueller [42]	2009	In-house	106	—	0.50±0.10	6.20±2.30	—	—	CBCT	—	ASM-based
Mannion-Haworth [43]	2015	PDDCA	48	92.67±1.00	—	—	1.98±0.59	—	CT	30 min/CPU	AAM-based
Babalola [44]	2009	MICCAI 2009	18	76.10±5.10 (exclude the 13th case)	—	—	—	—	CT	17 min/CPU	AAM-based

**Table 4 jpm-11-00629-t004:** Summary of atlas-based methods. ${}^{\left(1\right)}$ Values estimated from figures; actual values not reported.

Study	Year	Datasets	No. of Patients	Performance						Image Modalities	Time/ Equipment	Category
				Dice (%)	ASD (mm)	HD (mm)	95HD (mm)	VE (%)	Sen (%)/PPV (%)			
Chuang [45]	2019	In-house	54 + 20	97.60±10.60	—	—	—	—	—	CT	3–8 h/CPU	Atlas-based
Mencarelli [46]	2014	In-house	188	—	—	—	—	—	—	CT	—	Atlas-based
Chen [47]	2015	PDDCA	40	91.70±2.34	—	—	2.49±0.76	—	—	CT	100 min/CPU	Atlas-based
Han [48]	2008	In-house	10	90.00	—	—	—	—	—	CT	1 h/CPU	Atlas-based
Wang [49]	2014	In-house	13 + 30	91.00±2.00	0.61±0.17	0.92±0.47	—	—	—	CBCT + CT	—	Atlas-based
Zhang [50]	2007	In-house	7	80.00	0.85	—	—	—	—	CT	—	Atlas-based
Qazi [51]	2011	In-house	25	93.00	—	2.64	—	—	—	CT	12 min/CPU	Atlas-based
Gorthi [52]	2009	MICCAI 2009	18	77.80±7.40	—	16.87±6.75	—	—	—	CT	—	Atlas-based
Han [53]	2009	MICCAI 2009	18	90.33±1.49	—	8.07±3.12	—	—	—	CT	1 min/GPU	Atlas-based
Ayyalusamy [54]	2019	In-house	40	$85.00{\;}^{\left(1\right)}$	—	$9{\;}^{\left(1\right)}$	—	—	—	CT	—	Atlas-based
Haq [55]	2019	In-house PDDCA	45 32	$85.00{\;}^{\left(1\right)}$ $83.00{\;}^{\left(1\right)}$	—	—	$3.00{\;}^{\left(1\right)}$ $3.00{\;}^{\left(1\right)}$	$-5.00{\;}^{\left(1\right)}$ $52.00{\;}^{\left(1\right)}$	—	CT	—	Atlas-based
Liu [56]	2016	In-house	6	89.50	—	—	—	—	—	CT	10 min/—	Atlas-based
Zhu [57]	2013	In-house	32	89.00±4.00	—	9.80±4.10	—	—	—	CT	11.1 min/—	Atlas-based
Walker [58]	2014	In-house	40	98.00±2.00	—	—	—	—	—	CT	19.7 min/—	Atlas-based
McCarroll [59]	2018	In-house	128	84.00±7.00	1.89±1.55	18.63±14.90	—	—	—	CT	11.5 min/CPU	Atlas-based
La Macchia [60]	2012	In-house	5	89.00±2.00	—	—	—	−4.76±7.12	87.00±5.00/ 92.00±2.00	CT	10.6 min/CPU	Atlas-based
Zaffino [61]	2016	In-house	25	88.00±7.00	—	—	—	—	—	CT	120 min/CPU	Atlas-based
Huang [62]	2019	In-house	500	84.50±1.60	—	—	—	—	—	CT	—/GPU	Atlas-based

**Table 5 jpm-11-00629-t005:** Summary of level set-based methods.

Study	Year	Datasets	No. of Patients	Performance			Image Modalities	Time/ Equipment	Category
				Dice (%)	ASD (mm)	HD (mm)			
Wang [9]	2014	In-house	15	92.00±2.00	0.65±0.19	0.96±0.53	CBCT	5 h/CPU	Level set-based
Zhang [63]	2009	MICCAI 2009	18	87.93±2.06	—	8.70±3.28	CT	—	Level set-based

**Table 6 jpm-11-00629-t006:** Summary of classical machine learning-based methods.

Study	Year	Datasets	No. of Patients	Performance						Image Modalities	Time/ Equipment	Category
				Dice (%)	ASD (mm)	HD (mm)	Jac (%)	FPVF/FNVF (%)	MSE/RMSE (mm)			
Ji [64]	2013	In-house	12	97.90±1.10	0.20±0.13	—	95.80±2.00	—	—	MRI	128 s/CPU	CML
Wang [65]	2015	In-house	30 60	94.00±2.00 95.00±2.00	0.42±0.15 0.33±0.11	0.74±0.25 0.41±0.20	—	—	—	CBCT CT	20 min/—	CML
Udupa [66]	2014	In-house	15	—	—	3.30±0.56	—	1.00±0.00/ 49.00±8.00	—	MRI	54 s/CPU	CML
Linares [19]	2019	In-house	16	92.88	—	—	86.48	—	—	CBCT	5 min/CPU	CML
Barandiaran [67]	2009	In-house	12	—	—	—	—	—	—	CT	10 s/CPU	CML
Orbes-Arteaga [68]	2015	PDDCA	40	93.08±2.36	—	—	—	—	—	CT	—	CML
Wang [69]	2016	PDDCA	48	94.40±1.30	0.43±0.12	—	—	—	—	CT	—/CPU	CML
Qazi [70]	2010	In-house	25	90.19	—	—	—	—	—	CT	3 min/CPU	CML
Torosdagli [71]	2017	PDDCA	40	91.00	—	<1.00	—	—	—	CT	—/CPU	CML
Wu [72]	2018	In-house	216	89.00	—	1.60	—	—	—	CT	—	CML
Tam [36]	2018	In-house	56	85.20±5.30	—	—	—	—	0.10/3.16	CT	1 s/CPU	CML
Tong [73]	2018	In-house	246	—	—	—	—	—	—	CT	—	CML
Wu [74]	2019	In-house	216	89.00	—	—	—	—	—	CT	30 s/CPU	CML
Gacha [75]	2018	PDDCA	30	80.49	—	—	—	—	—	CT	—	CML
Spampinato [76]	2012	In-house	10	—	—	—	—	—	—	CT	—	CML

**Table 7 jpm-11-00629-t007:** Summary of deep learning-based methods.

Study	Year	Datasets	No. of Patients	Performance				Image Modalities	Time/ Equipment	Category
				Dice (%)	ASD (mm)	HD (mm)	IoU (%)			
Ibragimov [77]	2015	In-house	50	89.50±3.60	—	—	—	CT	4 min/GPU	DL
Kodym [78]	2019	PDDCA	35	94.60±0.70	0.29±0.03	—	—	CT	—	DL
Yan [79]	2018	In-house	93	90.76±2.45	—	—	85.44±3.99	CT	—/GPU	DL
Xue [80]	2021	PDDCA	48	94.00±2.00	0.49±0.18	2.36±0.62	—	CT	—/GPU	DL

**Table 8 jpm-11-00629-t008:** Summary of 3D network strategies of deep learning-based methods.

Study	Year	Datasets	No. of Patients	Performance							Image Modalities	Time/ Equipment	Category
				Dice (%)	ASD (mm)	HD (mm)	95HD (mm)	Rec/Sen (%)	RMSE (mm)	PPV (%)			
Chan [81]	2019	In-house	200	91.00±9.00	—	—	—	—	0.66±0.31	—	CT	20 s/GPU	DL
Zhu [82]	2019	PDDCA + TCIA	48 + 223	92.30	—	—	—	—	—	—	CT	0.12 s/GPU	DL
Xia [83]	2019	In-house	53	94.60±1.10	0.24±0.034	—	—	95.60±1.20	—	—	CT	1 s/GPU	DL
Willems [84]	2018	In-house	70	95.90	0.60	6.48	—	—	—	—	CT	1 min/GPU	DL
Ren [85]	2018	PDDCA	48	92.00;	—	—	1.89	—	—	—	CT	—/GPU	DL
He [86]	2020	StructSeg2019	50	90.30(L); 90.80(R)	—	—	—	—	—	—	CT	1 min/GPU	DL
Rhee [87]	2019	In-house + TCIA	1403 + 24	86.80±3.30	—	12.80±9.50	—	—	—	—	CT	2 min/GPU	DL
Nikolov [24]	2018	In-house TCIA PDDCA	459 30 15	93.10±1.90 92.90±3.50 93.80±1.90	—	—	—	—	—	—	CT	—/GPU	DL
Tong [88]	2019	PDDCA In-house	32 25	93.91±1.30 81.64±4.44	0.55±0.14 1.13±0.48	—	2.09±0.63 2.72±1.31	91.25±2.70 86.65±6.00	—	96.82±1.70 78.50±4.30	CT MRI	14 s/GPU	DL
Xue [89]	2019	PDDCA	48	95.70±1.80	—	—	0.60±0.49	—	—	—	CT	—/GPU	DL

**Table 9 jpm-11-00629-t009:** Summary of 2.5D networks deep learning-based methods.

Study	Year	Datasets	No. of Patients	Performance				Image Modalities	Time/ Equipment	Category
				Dice (%)	ASD (mm)	95HD (mm)	RMSE (mm)			
Qiu [5]	2019	In-house PDDCA	109 40	88.10 93.28±1.44	—	— 1.43±0.56	0.58 —	CT	2.5 min/GPU	DL
Lei [90]	2021	StructSeg2019	50	91.10±2.90(L); 91.70±1.50(R)	—	2.81±0.45(L); 2.70±0.40(R)	—	CT	2 min/GPU	DL
		StructSeg2019 + PDDCA + In-house	50 + 48 + 67	90.00±4.20		6.54±19.14				
Liang [91]	2020	PDDCA In-house	48 96	94.10±0.70 91.10±1.00(L); 91.40±2.00(R)	0.28±0.14 0.76±0.13(L); 0.86±0.14(R)	—	—	CT	—/GPU	DL
Qiu [92]	2020	In-house PDDCA	109 40	97.53±1.65 95.10±1.21	0.21±0.26 0.14±0.04	2.40±4.61 1.36±0.45	—	CT	1.5 min/GPU	DL

**Table 10 jpm-11-00629-t010:** Summary of attention strategies of deep learning-based methods.

Study	Year	Datasets	No. of Patients	Performance					Image Modalities	Time/ Equipment	Category
				Dice (%)	ASD (mm)	95HD (mm)	Rec/Sen (%)	PPV (%)			
Gou [93]	2020	PDDCA	48	94.00±1.00	0.47±0.11	1.40±0.02	93.00±2.00	95.00±2.00	CT	2 s/GPU	DL
Jiang [94]	2019	In-house + PDDCA	48 + 48	93.00±1.00 (trained in in-house dataset)	—	—	—	—	CT	0.1 s/GPU	DL
Sun [95]	2020	In-house	129	94.05±1.08	—	—	—	—	CT	—/GPU	DL
Liu [96]	2020	StructSeg2019	50	90.28(L); 90.81(R)	—	—	—	—	CT	—/GPU	DL

**Table 11 jpm-11-00629-t011:** Summary of two-stage strategies in deep learning-based methods.

Study	Year	Datasets	No. of Patients	Performance			Image Modalities	Time/ Equipment	Category
				Dice (%)	HD (mm)	95HD (mm)			
Zhang [97]	2021	In-house	170	89.00±2.00	—	1.66±0.51	CT	40.1 s/GPU	DL
Tappeiner [98]	2019	PDDCA	40	91.00±2.00	—	2.4±0.6	CT	38.3 s/GPU	DL
Mu [99]	2020	In-house	50	89.80±2.70(L); 90.40±2.00(R)	—	—	CT	3 s/GPU	DL
Wang [100]	2018	PDDCA	48	93.00±1.90	—	1.26±0.50	CT	6 s/GPU	DL
Tang [32]	2019	In-house HNC+ HNPETCT PDDCA	175 35 + 105 48	93.12±1.41 89.31±11.59 95.00±0.80	—	2.48±0.83 3.05±2.60 —	CT	2 s/GPU	DL
Lei [101]	2020	In-house	15	85.00±4.00	—	—	MRI	—	DL
Lei [102]	2020	In-house	15	88.00±3.00	—	—	CT	—	DL
Liang [103]	2019	In-house	185	91.40±0.04(L); 91.20±3.00(R)	—	—	CT	30 s/GPU	DL
Dijk [104]	2020	In-house	693	94.00±1.00	—	1.30±0.50	CT	—	DL
Men [105]	2019	HNSCC	100	92.00±2.00	2.40±0.40	—	CT	5.5 min/GPU	DL
Egger [106]	2018	In-house	20	89.64±1.69	—	—	CT	—/CPU	DL

## Data Availability

All relevant data are included in the study.

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
