# Peer review of "Automatic Segmentation of Mandible from Conventional Methods to Deep Learning—A Review"

_jpm, 2021, doi:10.3390/jpm11070629_

Round 1
Reviewer 1 Report
The title “Automatic segmentation of mandible from conventional methods to deep learning - A review.” is appropriate and clear.
The paper presents the available fully (semi)automatic segmentation methods of the mandible published in different scientific articles. This review provides a vivid description of the scientific advancements to clinicians and researchers in this field to help develop novel automatic methods for clinical applications. The review is generally very good, with limited description of the segmentation models on PET/CT. Therefore, my recommendation is accepted after ‘Minor Revision’. More detailed comments are given below.
- Pag 4. In section “Image modality” the authors speak about "While CT images provide a good visibility of the bony anatomy, the contrast differences between various soft tissues are much lower than those in MRI. In the consensus on manual delineation of the mandible based on CT, it is also necessary to use MR images, in addition to CT, to help delineate surrounding tumors and soft tissues. Furthermore, considering the ionizing radiation of CT/CBCT scanning and the development of computer vision techniques, a strategy of MRI-only treatment planning has become increasingly valuable and feasible in the field of 3D VSP[3]." without referring to a very important method for delineating H&N lesions such as PET / CT.
PET/CT for radiation treatment planning in head and neck cancer (H&N) is extremely useful in:
Staging patients with H&N cancer, since it can alter the TNM stage up to 30% of patients
Delineation of the target is more accurate than CT/MRI in the definition of GTV and reduces interobserver variability
Dose escalation
Treatment planning
Treatment delivery
Evaluation of the response and followup
Biological characterization of the disease
Assessment of treatment response
PET/CT has a sensitivity of 98%, a specificity of 92% and an accuracy of 92%, compared with 74%, 75% and 74% of CT scan respectively (Branstetter, Radiology 2005)
deserves an in-depth study and the inclusion of the method which, in addition to providing morphological information with CT, also gives metabolic information(Biological characterization) with PET. - 5 : In subsection “3.3. Evaluation metrics” the authors did not report in the tables of the important performance parameters that they mention in the intro but which they do not evaluate in the tables such as:
the time taken to make a inference using of the model.
the computational complexity and performance of the model (Number of Parameters "Trainable and Non-Trainable", Size on disk and Inference Times / dataset on CPU and GPU).
model training Times / dataset on CPU and GPU).
Please improve these points.
Pag.1-2 and Pag.5-15: In section “1. Introduction” and subsection “3.4. Methodology” the literature review, although excellent, could be refer to works about fully (semi)automatic segmentation using hybrid active contour models 3D applied to PET/CT such as:
Fully 3D Active Surface with Machine Learning for PET Image Segmentation. J. Imaging 2020, 6, 113. https://doi.org/10.3390/jimaging6110113.
In this paper the author present an algorithm capable of achieving the volume reconstruction directly in 3D, by leveraging an active surface algorithm. The evolution of such surface performs the segmentation of the whole stack of slices simultaneously and can handle changes in topology. Furthermore, no artificial stop condition is required, as the active surface will naturally converge to a stable topology. In addition, I include a machine learning component to enhance the accuracy of the segmentation process. The latter consists of a forcing term based on classification results from a discriminant analysis algorithm, which is included directly in the mathematical formulation of the energy function driving surface evolution. It is worth noting that the training of such a component requires minimal data compared to more involved deep learning methods. Only eight patients (i.e., two lung, four head and neck, and two brain cancers) were used for training and testing the machine learning component, while fifty patients (i.e., 10 lung, 25 head and neck, and 15 brain cancers) were used to test the full 3D reconstruction algorithm. The results confirm that the active surface algorithm is superior to the active contour algorithm, outperforming the earlier approach on all the investigated anatomical districts with a dice similarity coefficient of 90.47 ± 2.36% for lung cancer, 88.30 ± 2.89% for head and neck cancer, and 90.29 ± 2.52% for brain cancer.
Development of a new fully three-dimensional methodology for tumours delineation in functional images. Computers in biology and medicine, 2020, 120: 103701. https://doi.org/10.1016/j.compbiomed.2020.103701
Please integrate these informations.
Reviewer 2 Report
This literature review gives a good overview of mandibular segmentation techniques. Indeed, with extended metallic reconstructions, artefact formation is a problem. There are imaging-specific algorithms and acquisition techniques that can optimise this in some ways, but cannot eliminate them 100%. The review deals with the technical aspects less whether it is clinically relevant (partly mentioned in the discussion). In the introduction, mandibular segmentation is described as difficult and time-consuming. I do not fully agree with this, I think you also have to look at it from a clinical point of view, especially as mandibular segmentations can be performed quickly for daily clinical use, even with open-source software. There are other anatomical structures that are much more difficult and time-consuming to segment. In my opinion, this should be adjusted.
